# Reimagining Microbially Induced Concrete Deterioration: A Novel Approach Through Coupled Confocal Laser Scanning Microscope–Avizo Three-Dimensional Modeling of Biofilms

**DOI:** 10.3390/microorganisms13071452

**Published:** 2025-06-23

**Authors:** Mingyue Ma, Guangda Yu, Zhen Xu, Jun Hu, Ziyuan Ji, Zihan Yang, Yumeng Sun, Yeqian Zhen, Jingya Zhou

**Affiliations:** 1Department of Agricultural Resources and Environment, Yanbian University, Yanji 133002, China; 19889738736@163.com (M.M.); 15615712036@163.com (Y.Z.); 2Department of Environmental Science, Yanbian University, Yanji 133002, China; guangdayu_zju@163.com (G.Y.); jzy1878026861@163.com (Z.J.); 15643831183@163.com (Z.Y.); 15948095767@163.com (Y.S.); 3College of Geography and Ocean Science, Yanbian University, Yanji 133002, China; xuzhen@ybu.edu.cn; 4School of Civil Engineering and Architecture, Hainan University, Haikou 570228, China; hj7140477@hainu.edu.cn

**Keywords:** biofilm, Avizo, three-dimensional modeling, MID, corrode

## Abstract

Microbially induced concrete deterioration (MID) poses a significant and urgent challenge to urban sewerage systems globally, particularly in tropical coastal regions. Despite the acknowledged importance of biofilms in MICC, limited research on sewer pipe biofilms has hindered a comprehensive understanding of their deterioration mechanisms. To overcome this limitation, our research employed multiple staining techniques and digital volume correlation (DVC) technology, creating a new method to analyze the microstructure of biofilms, precisely identify the components of EPSs, and quantitatively examine MID mechanisms from a microscopic viewpoint. Our results revealed that the biofilm on concrete surfaces regulates the types of amino acids, thereby creating an environment conducive to microbial aggregate survival. Additionally, salinity significantly influences biofilm component distribution, while proteins play a pivotal role in biofilm mechanical stability. Notably, a high salinity fosters microbial migration within the biofilm, exacerbating deterioration. Through this multidimensional inquiry, our study established an advanced echelon of comprehension concerning the intricate mechanisms underpinning MICC. Meanwhile, by peering into the biofilms and elucidating their interplay with concrete, our findings offer profound insights, which can aid in devising strategies to counter urban sewer system deterioration.

## 1. Introduction

In modern societies, efficient, safe, and cost-effective sewerage networks play a vital role in safeguarding public health and preventing environmental pollution (Anbari et al., 2017) [1]. As temenoses attach directly to the surface of concrete materials, their components and structure are closely related to changes in the surface material. Moreover, anaerobic microbial communities in the temenos perform redox reactions under anaerobic conditions, generating biogenic sulfuric acid, which leads to concrete deterioration. High temperatures, high salt levels, and exposure to coastal seawater pose serious deterioration risks for these essential infrastructure systems. Under such harsh conditions, urban sewage pipelines become highly vulnerable to deterioration and premature failure. To enhance the performance of concrete structures in marine engineering, it is imperative to account for the impact of temenos adhesion on the deterioration of concrete materials (Grengg et al., 2018; Li et al., 2022) [2,3]. Previous studies have investigated concrete samples that exhibit significant deterioration damage, revealing evidence of temenos adhesion and penetration (Gaylarde & Ortega-Morales, 2023) [4]. Notably, the corrosive influence of acidic Thiobacillus on post-deterioration irregular surfaces has been observed (Kip & Veen, 2014; Zhou et al., 2021) [5,6]. Microbial cells adhere to marine concrete surfaces, creating salt-resistant temenoses (Xie et al., 2019) [7]. This process includes extracellular proteins, nucleic acids, and lipids produced by salt-tolerant microorganisms, promoting the formation of the adhesive layer. Thus, examining the internal structure of temenoses and the functional roles of their individual components is essential for understanding the processes by which temenoses corrode concrete materials.

Additionally, the composition of EPSs significantly influences the role of microbial aggregates. Research indicates that the main components of EPSs are proteins, polysaccharides, nucleic acids, and lipids (Windt & Devillers, 2010; Wingender et al., 1999) [8,9]. Each component has distinct functions, which collectively determine the characteristics and efficacy of EPSs and microbial aggregates (Grengg et al., 2018; X. Li et al., 2022) [2,3]. In particular, extracellular proteins in microbial aggregates primarily assist in the binding of multivalent cations and organic molecules, which facilitates the catalysis and degradation of large organic molecules (Zhu et al., 2015) [10]. Zhang identified and performed a detailed structural analysis of the authentic flocculent structure within these aggregates, establishing a comprehensive flocculation model (Zhang & Bishop, 2001) [11]. Later studies further confirmed that temenos performance is influenced not only by temenos thickness, but also by its functional three-dimensional internal structure (Dreszer et al., 2014; West et al., 2016) [12,13]. The structure of EPSs within temenoses can alter transport functions, significantly impacting attributes such as the porosity and permeability (VandeWalle et al., 2012) [14]. Nonetheless, past research has not explored how microorganisms adhere to concrete surfaces to form temenoses, or how they sustain themselves with nutrients and migrate to infiltrate the internal structure of the corroded concrete layer.

For a more effective management of MID, a comprehensive grasp of temenos structures within sewer systems is imperative. However, our understanding of temenos structures within anaerobic sewer systems in high-temperature, high-salinity environments remains limited. The Avizo software (AVIZO 9.1) allows for the three-dimensional visualization and analysis of complex structures (Ennazii et al., 2024) [15], presenting the 3D structure of key EPS components such as proteins and polysaccharides at the micron scale. Layout algorithms like the Fruchterman–Reingold, distance geometry, and hierarchical matter layouts can be used to arrange these components. Data on these layers can be clustered using a k-means analysis, and a representative elementary volume analysis helps optimize the image resolution parameters to characterize the temenoses (Chrostek et al., 2024) [16]. Consequently, we utilized the Avizo 3D software(AVIZO 9.1) to analyze confocal laser scanning microscope (CLSM) images with multiple fluorescent labels, generating a 3D visualization of the EPS structure. This comprehensive view of the three-dimensional temenos architecture unveiled the interactions between concrete materials and temenoses under adverse conditions. Furthermore, this study examined the dynamic responses of microbial aggregates and quantitatively analyzed individual components within the temenos, thereby elucidating the MID process.

## 2. Materials and Methods

### 2.1. Preparation of Concrete Material Specimens

Type II ordinary Portland cement 42.5R (C), ground granulated blast-furnace slag (GBFS), basalt powder (BP), desulfurized gypsum (DG), and mixed coarse aggregate basalt (CAB) with particle gradations of 3–5 mm and 5–10 mm were employed in this study. The sand, which was locally sourced in Hainan, featured a fineness modulus of 2.62 (Z. Li et al., 2020; Sand & Gehrke, 2006) [17,18]. To enhance the workability, a polycarboxylate-based superplasticizer (HR) was incorporated into the concrete mixtures, resulting in a blending ratio for C, GBFS, BP, DG, HR, sand, and CAB of 1:0.85:0.22:0.04:3.26:0.04. Furthermore, the water–cement ratio was established at 0.38. Detailed information regarding the chemical composition and physical properties of C, GBFS, BP, and DG can be found in Appendix A. On the other hand, the HCS 60 MPa standard (JGJ55-2011) (China Academy of Building Research) was formulated based on the mixture proportions of conventional concrete samples [19]. These standard concrete specimens, measuring 160 × 40 × 40 mm, underwent preparation (Zhou et al., 2021) [6]. Subsequently, they were subjected to curing intervals of 3, 7, and 28 days under controlled conditions of 20 °C and 100% relative humidity. After this curing period, the compressive and flexural strength of the specimens was evaluated and the results are shown in Appendix A. Following the standardized 28-day curing duration, a paraffin coating was selectively applied to the surfaces of the samples, reserving a specific 4 × 16 cm^2^ side for subsequent deterioration assessments.

### 2.2. Design and Operating Conditions of Anaerobic Concrete Pipeline

This study included the independent design and assembly of six parallel operating anaerobic pipelines with a volume of 76.93 L per pipeline. These pipelines were constructed using concrete material as the base material due to its compact structure and low porosity, ensuring the reactor’s stable operation over an extended period. Each pipeline was designed with a cylindrical single wall, measuring 80 cm in outer diameter, 70 cm in inner diameter, and 20 cm in height, as shown in Figure 1.

At present, most of the laboratory simulation studies of the environmental conditions in the pipeline basically alternate between aerobic and anaerobic conditions. However, due to the presence of sewage sediments in the actual concrete sewage pipeline, the dissolved oxygen level in the sewage is close to anaerobic (VandeWalle et al., 2012) [14]. Therefore, in this study, the pipeline was covered with a certain thickness of organic glass, and the contact area was filled with glass glue to maintain a sealed anaerobic environment, so as to more realistically complete the simulation of an urban sewage pipeline network system.

The simulation was conducted to account for the unique environmental conditions of coastal cities, with a comparison to a freshwater area. Consequently, this study set the salinity to 35 g/L, 18.95 g/L, and 3 g/L and a controllable temperature heater was placed in the reactor to maintain the temperature of the reactor at 35 ± 2 °C and 25 ± 2 °C to encompass the typical temperatures of coastal cities. This approach allowed for a comprehensive investigation of microbial activities under different salinity conditions, providing valuable insights into deterioration management and sewage system optimization in coastal areas.

### 2.3. Inoculation and Domestication of Activated Sludge

The sludge used in this study for inoculation in the anaerobic pipeline reactor was obtained from the Baishamen Wastewater Treatment Plant in Haikou, Hainan Province, China. The wastewater treatment plant is located at the northwest end of Haidian Island, Haikou City, approximately 0.7 km north of the existing seawall and on reclaimed land west of Baishajiao. It processes 300,000 tons of sewage from the main urban area of Haikou daily, and the wastewater treatment plant was operating steadily before the sampling.

Salinity was introduced into the experimental group, following the elemental composition of seawater. In order to maximize the efficiency of the reactor operation, we needed to make the sulfate-reducing bacteria inside the reactor an absolute advantage. Subsequently, we calculated the concentration of magnesium sulfate and sodium chloride required to be added based on the absolute advantage of sulfate-reducing bacteria in the anaerobic flora when the COD/sulfate = 1:1 in previous studies (Balemans et al., 2020) [20].

### 2.4. Monitoring of Chemical Parameters Inside the Pipeline

In this study, the sewage samples in the anaerobic pipeline were collected by the self-made sampling device of the reactor. The collection cycle lasted for 7 days, and the data of 3 cycles were collected. A pH electrode (E-201-C) was used to monitor and collect the pH value of the sewage in the anaerobic pipeline in real time, and the COD of the sewage was determined by the potassium dichromate method. A hydrogen sulfide electrochemical sensor (SOLIDSENSE 4 H2S-1000, Stuttgart, Germany) and a model (CLE-0113-400) H2S probe were used to collect the gas environment in the pipeline reactor, and the multi-channel data transmission was controlled by programming. The change in hydrogen sulfide gas in the reactor under different external environment conditions was detected at the same time, and the purpose of multi-channel real-time data acquisition was realized.

### 2.5. Temenos Collection and Measurement of EPS Component Content

Temenos samples were scraped from the surfaces of three areas with the most severe concrete deterioration in both the low-salt and high-salt groups using sampling knives. The samples were mixed, placed in sterile bags, and stored at −20 °C. The extraction of EPS from the temenos mark was performed using the formaldehyde-sodium hydroxide method. Firstly, activated sludge samples were placed in centrifuge tubes, and 50 mL of PBS buffer was added. Next, 0.06 mL of a 36.5% formaldehyde solution was added, resulting in a final concentration of 0.044% (*v*/*v*) formaldehyde in the PBS buffer. The centrifuge tubes were incubated at 4 °C for 1 h with gentle shaking (800 r/min) to allow complete fixation. Then, 4 mL of a 1 mol/L NaOH solution was added, and the mixture was incubated at 4 °C for 3 h with intermittent shaking (every 30 min) to facilitate EPS extraction. After incubation, the samples were centrifuged at 800 r/min for 10 min to separate the supernatant. The supernatant was collected and the samples were freeze-dried for 12 h to obtain a powder-like form, which was stored in a −80 °C ultra-low-temperature freezer for later use. Then, the protein in the EPS was determined by Folin’s phenol reagent method, and the polysaccharide content in the EPS was determined by the phenol–sulfuric acid method. Finally, SEM temenos observations and an Avizo visualization analysis were carried out with a temperature of 35 °C and a salinity of 3 g/L (FQHC) or 35 g/L (SQHC) as representatives of tropical coastal areas and freshwater areas to determine the effect of salinity on the temenos structure.

### 2.6. Determination of Protein and Polysaccharide Content in EPS

Folin’s phenol reagent method is easy to execute and can be used to detect trace proteins. It is one of the most sensitive protein determination methods and has a high accuracy. It has been widely used in the field of biochemistry. The protein in the EPS extracted in this study was a trace protein, so Folin’s phenol reagent method was used for detection. At the same time, the content of polysaccharides in the EPS was determined by the phenol–sulfuric acid method.

### 2.7. SEM Observation

In order to minimize the changes in the composition of the temenos caused by drying, the corroded concrete samples were taken from the inside of the anaerobic concrete reactor and stored in an ice box in time. They were then freeze-dried for 24 h. Next, gold was sprayed for 200 s, and a scanning electron microscope (SEM) (TESCAN MIRA3 and Thermo Fisher NS7, Waltham, MA, USA) was used for the analysis. Paste specimens with dimensions of 1 cm × 1 cm × 1 cm were extracted from the reactor. To facilitate the observation of the deterioration of materials, we utilized a diamond wire cutting machine (STX-202A, KEJING, Shenyang, China) to precisely cut and sample. The samples were dried at 60 °C for 12 h. The positions of different layers on the surface (0–1 mm) and/or within (1–2 mm) the specimens were selected for observation, as shown in Figure 2. Before the SEM observations, the samples were sprayed with gold for 60 s prior to testing. The micro-morphologies of the samples were observed using an SEM (S4800, Hitachi, Tokyo, Japan) operating at 10.0 kV or an SEM (MIRA3, Tescan, Brno, Czech Republic) operating at 5.0 kV. Additionally, energy-dispersive X-ray spectroscopy (EDS) was performed in conjunction with SEM to analyze the elemental composition and distribution of the corroded materials. This technique provided atomic-level insights into the chemical changes induced by microbial activity, particularly in identifying sulfur (S), calcium (Ca), iron (Fe), and other key elements associated with concrete deterioration and biofilm formation (Wang J, Yin S, Lu L, et al., 2022) [21].

### 2.8. EPS Fluorescent Staining and CLSM Observation

To directly visualize the distribution of EPSs throughout the deteriorated concrete sample, a mixture of five dyes was used. The dyeing solution and its related parameters are provided below. PI was used to stain dead bacteria, rhodamine was used to stain all bacteria, Nile red was used to stain the lipids, optical brighteners were used for the polysaccharides, and FITC was used for the proteins, as shown in Table 1. Meanwhile, a multi-staining scheme was adopted; in order to facilitate the differentiation of the components in the temenos, the proteins, polysaccharides, and nucleic acids were mixed and the fat and bacteria were single-stained, so that each component could be visualized in a certain thickness of the temenos. The structure and distribution of the temenos, which was composed of bacteria and extracellular polymers, were examined using an LEICATCS SP8 CLSM (Wetzlar, Germany).

### 2.9. Determination of Fluorescence Spectroscopy

A three-dimensional excitation–emission matrix (3D-EEM) was used to determine the fluorescent substances in the EPS extracted from the temenos (Pan et al., 2010) [22]. The freeze-dried EPS sample powder was prepared at 1 mg/mL and immediately placed in a fluorescence spectrophotometer (F-7000 FL, Hitachi, Japan) instrument to determine its fluorescence spectrum. The excitation wavelength of the fluorescence spectrophotometer was set between 200 nm and 400 nm, and the sampling interval was 5 nm. At the same time, the emission spectrum increment from 220 nm to 500 nm was set to 5 nm, and the scanning speed was set to 3000 nm/min.

### 2.10. Avizo Modeling

After using CLSM imaging technology to locate and quantitatively analyze EPSs in the sewage pipeline system (Chen et al., 2007) [23], combined with modules in the Avizo software (AVIZO 9.1), such as threshold segmentation, binarization processing, and volume extraction, the appropriate threshold was selected to obtain fine structural and quantitative data for the temenos. The analysis process primarily included four steps: shadow correction, grayscale image conversion, image segmentation, and target measurement.

The microbial aggregates and EPS in the temenos were layered along the thickness of the temenos, and the production of the microbial aggregates and EPS also varied with the thickness of the temenos (Zhang & Bishop, 2001) [11]. Therefore, the Avizo software (AVIZO 9.1) was used to statistically analyze the images of various components (fluorescent volumes) in the temenos. In the case of a known temenos sample quality, the overall mass of the different components in the temenos can be calculated according to Formulas (1) and (2). The adopted method was relatively compared to the colorimetric determination of the component content, avoiding the influence of different environmental conditions and human errors, and this method was also used to calculate the total EPS content.(1)$${EPS}_{I}=\frac{m}{v}V_{I}$$(2)$${EPS}_{Ⅱ}=\frac{m}{V}(V-V_{B})$$

In the equation, m represents the mass of the temenos; V represents the volume of the temenos; EPS_I_ represents the content of proteins, polysaccharides, or lipids; V_I_ represents the volume of proteins, polysaccharides, or lipids; EPS_II_ represents the total mass of the EPS; and V_B_ represents the volume of bacteria.

## 3. Results and Discussion

### 3.1. Analysis of Chemical Parameters of Concrete Pipeline Reactor

Studies have shown that concrete deterioration is related to microbial metabolism, such as sulfur-oxidizing bacteria (SOB) and sulfate-reducing bacteria (SRB). At the same time, the chemical transformation of microorganisms on the surface of temenoses plays a key role in regulating the temenos’s physiological functions (Wu et al., 2020) [24]. The bio-acids formed by bacterial metabolism will corrode the concrete surface they adhere to and change the pH value of the concrete surface. Therefore, understanding the changes in the chemical parameters of wastewater can reflect the growth of microorganisms and further predict the deterioration of sewage pipelines (Liu et al., 2023) [25]. After three cycles in the anaerobic concrete pipeline reactor, the COD levels in the wastewater showed a downward trend (Appendix A), while the pH levels trended upward (Appendix A). This observation aligns with previous studies (Brand et al., 2014; Brisolara & Qi, 2013) [26,27], suggesting that the pipeline’s internal environment has reached stability, with SRB and SOB likely being the dominant microbial species within the pipeline. The decrease in the COD levels in wastewater within anaerobic pipelines is similar in low- and medium-salinity environments, with little variation observed across different temperatures. However, in high-salinity environments, the decrease in the COD is less significant, and this reduction further diminishes as the temperature increases. This suggests that only when salinity reaches a certain range does the COD decrease become affected by both salinity and temperature. To gain a more precise understanding of the internal reactions and mechanisms within the anaerobic pipeline reactor, a further analysis of wastewater parameters is required.

Research shows that the pH is an essential factor affecting the growth and reproduction of SRB, with a growth pH range of 5.0 to 9.0, and with SRB activity being inhibited at pH levels above 8.6 (Gutierrez et al., 2009; Hao et al., 2014) [28,29]. In this study, the pH within the various anaerobic pipeline reactors ranged approximately from 7 to 8.3, showing some fluctuation and a general upward trend. This shows that SRB in the anaerobic concrete pipeline reactors maintained a high level of activity, and that the internal reactions within the pipeline were complex and dynamically balanced. Furthermore, during the reaction, the pH of the wastewater gradually increased, with the pH under high-salinity conditions consistently higher than that under low-salinity conditions. The internal sulfate reduction reaction in the wastewater was acidic in nature. As the reaction progressed, significant H_2_S was produced in high-salinity pipelines (Appendix A), continuously providing sulfur compounds to SOB on the corroded sample surfaces and leading to severe deterioration. The higher the salinity, the greater the production of biogenic sulfuric acid, resulting in the more severe deterioration of concrete materials. This ongoing reaction led to a decrease in the H_2_S gas content under medium- and high-salinity conditions (Appendix A), although this trend was less noticeable under low-salinity conditions due to the lower initial H_2_S gas levels. The dissolved slurry and powder from the corroded concrete further increased the wastewater’s pH. The increased stability of the pH and the higher H_2_S levels at elevated temperatures could be attributed to the fact that SRB, as thermophilic organisms, have optimal growth and metabolic activity under high-temperature conditions, leading to a sustained high rate of sulfate reduction (Diao et al., 2023) [30].

### 3.2. Microbial Community Composition and Diversity

This study compared and analyzed reactors under different salinities, and the 16S ribosomal ribonucleic acid gene sequence method was used to analyze the bacterial communities associated with concrete corrosion. A redundancy analysis demonstrated that the chemical oxygen demand, volatile fatty acids, and sulfate altered the structure and distribution of the microbial community. The predominant bacteria, Proteobacteria, accounted for 41.85% of the seawater group. Among them, the functional sulfur microorganisms, Desulfomicrobium, accounted for 4.14%. These bacteria can decompose macromolecular organic matter to provide energy for reproduction. Furthermore, they continued to provide sulfur for the eosinophilic sulfur-oxidizing bacteria attached to the surface of the high-alkaline concrete sample. The aggregated sulfur-oxidizing bacteria produced biological sulfuric acid, leading to corrosion and damage to the concrete structure. Salinity promoted the aggregation of corrosion-inducing bacteria, accelerating the growth of corroding microorganisms on the concrete material of coastal urban sewage pipes (Zhou J, Yin S, Fu Q, et al., 2021) [6].

### 3.3. Components and Physicochemical Properties of Temenoses

Polysaccharides and proteins constitute the primary components of temenoses (Kong et al., 2022) [31]. In this study, the protein and polysaccharide contents within temenoses were assessed, revealing that the polysaccharide levels consistently surpassed those of proteins under varying temperature and salinity conditions (Appendix A), aligning with Hui et al.’s findings on SOB temenoses (Rong et al., 2021) [32]. However, temenoses in natural environments are usually composed of multiple microorganisms, complicating the situation. Given that polysaccharides are key to enhancing EPS participation in surface adhesion and ensuring temenos structural coherence (Annuk & Moran, 2010) [33], further investigation is needed. Furthermore, high-salinity conditions not only diminish microbial activity, but also hinder the proper expression of certain proteins. With increasing salinity, protein levels within temenoses decline gradually (Appendix A). Studies show that microbial richness and community diversity are lower in high-salinity pipelines compared to low-salinity environments (Zhou et al., 2021) [6], which results in a reduced protein content as salinity increases and a corresponding decrease in functional proteins within temenoses in high-salinity settings. Additionally, because microbial communities in high-temperature, high-salinity environments are more specialized, the protein expression exhibits a greater specificity (Meng et al., 2018) [34].

The 3D-EEM spectroscopic results of temenoses, as depicted in Figure 2, clearly demonstrate the presence of two distinct 3D-EEM spectral peaks—Peak A (Ex/Em: 275–280/325–350 nm) and Peak B (Ex/Em: 225/300 nm) in EPSs under varying salinity conditions, indicating the presence of tryptophan and tyrosine in this substance (Han et al., 2023; Su et al., 2023) [35,36]. Additionally, with increasing salinity, the intensities of Peaks A and B gradually decreased. It is evident that, in low-salinity environments, the levels of tyrosine and tryptophan are higher than in high-salinity environments. Research has indicated that, although tyrosine and tryptophan do not impede bacterial growth, they diminish the adhesion efficiency of the temenos and suppress polysaccharide secretion, particularly during the initial stages (Kolodkin-Gal et al., 2010; Su et al., 2021) [37,38] This suggests that, in low-salinity environments, bacteria produce these two amino acids to restrain bacterial aggregation and surface adhesion (H. Li et al., 2023) [39], thereby facilitating the expansion and migration of the temenos, enabling more nutrients to be obtained, and causing a broader deterioration effect. Conversely, under the stress of high salinity, bacteria reduce the production of these two amino acids to maintain the stability of the temenos structure, ensuring the survival of bacterial populations in the harsh high-salinity environment. Furthermore, a high salinity can lead to disturbances in the amino acid metabolism of microorganisms, and shifts in the dominant microbial communities can also result in changes to the physicochemical characteristics of temenoses (J. Li et al., 2018) [40].

### 3.4. SEM Image Analysis

The SEM observations depict the characteristics of the temenos in different salinity environments. The observations reveal a substantial coverage of the temenos with a high bacterial density (Figure 3a). Furthermore, certain regions on the surface of the temenos exhibited uneven structures, and there was evidence of cell separation leading to the formation of new spores. Previous research suggests that these new spores are predominantly arc-shaped bacteria, tentatively identified as desulfurizing arc bacteria (Zhou et al., 2021) [6], at the stage of spore coagulation and EPS attachment. Upon further magnification, it was observed that some newly formed spores were attached to the EPS on the concrete surface (Figure 3a). Concurrently, the microbial community adhering to the surface initiates proliferation and gradual aggregation, resulting in an augmentation of the overall density and complexity of the temenos, culminating in the formation of a glycocalyx.

Within a high-salinity environment, mature temenoses undergo numerous biological processes, including quorum sensing, gene transfer, and persistent development. Notably, under high-salinity conditions, temenoses aggregate visibly, and newly formed spores disperse deeper into the concrete from the matrix voids (Figure 3b), which aligns with the protein and polysaccharide distribution determined in the subsequent section using the Avizo software (AVIZO 9.1) analysis. The separation of temenoses and the release of free bacteria aid in both the survival of microbial aggregates within the temenos and the accomplishment of the temenos colonization process (Kaplan et al., 2003) [41].

In addition, in SQHC, the production of gypsum and ettringite exerts significant damage pressure on the microstructure, leading to the expansion of surface pores in the concrete (Figure 4a). Concurrently, metabolic reactions occurring within the temenos of anaerobic sewage pipeline reactors continually generate biogenic acids. These biogenic acids gradually erode the calcium hydroxide structure (Figure 4b) with the aid of the gel-pore network formed, further enhancing the permeability of concrete. This leads to the deeper degradation of concrete while providing attachment sites for microorganisms (Figure 4c). As the penetration process deepens, it subsequently induces the formation of expansive microcracks within the material, the partial separation of the temenos, and its continuous diffusion into the cracks of the matrix (Figure 4d), ultimately resulting in severe internal material deterioration.

### 3.5. Three-Dimensional Reconstruction of Temenos

The three-dimensional structure of the temenos revealed three-dimensional correlations between fluorescent substances within the scanned object, as depicted in Figure 5. In low-salinity environments, the temenos’s components were concentrated in the central region, and the distribution of these components was relatively uniform. Conversely, in high-salinity environments, the various components exhibited an irregular distribution within the temenos model area and tended to distribute in deeper layers of the temenos model. Additionally, the proportion of polysaccharides located in the deeper layers was relatively high and they remained evenly distributed. Simultaneously, the elevated concentration of polysaccharides enhanced the cohesiveness of the temenos (Ahimou et al., 2007) [42]. Meanwhile, proteins tended to cluster at the edges of the temenos, forming a highly organized structure that enabled microbial aggregates to survive on the surface of the concrete pipes in a high-alkaline environment. The results obtained from this model align with the SEM analysis of the high-salinity temenos samples mentioned earlier, where new spores continued to diffuse into matrix pores and secrete organic substances through metabolic processes, leading to further severe material deterioration. A more specific demonstration of the 3D model structure is detailed in Appendix A (FQHC) and Appendix A (SQHC).

Apart from the impact of the EPS composition and distribution, the temenos also displayed a highly layered and heterogeneous structure in depth (Ahimou et al., 2007) [42]. An analysis of the volume data of different EPS components extracted by the Avizo software (AVIZO 9.1) enabled us to determine the proportion of EPS components at various depths of the temenos adhering to corroded concrete specimens, as illustrated in Figure 6.

The composition analysis of the temenos showed that the distribution of each component in the low-salt temenos sample tended to be consistent, and was roughly distributed in the center of the temenos. The results displayed by the model were consistent with the results observed in the above SEM. In the low-salinity environment, the temenos samples adhered together to form an independent and complete polymer, with only a small number of pores, that is, a small number of new spores gradually migrated, resulting in the separation and transfer of temenos parts. With an increase in the temenos depth, the distribution of various components in the temenos also increased (Hartmann et al., 2018) [43], but not all the components were evenly distributed along the temenos depth. However, in the high-salt temenos samples, the depth positions of various components were inconsistent; in particular, the protein tended to be distributed in the deeper layer of the temenos model. The results obtained by this model are still consistent with the above SEM analysis of high-salt temenos samples. The new spores continued to diffuse into the matrix pores, resulting in the further severe deterioration of the material. In addition, the proportion of proteins and polysaccharides in the deep layer was significantly higher in the high-salt environment than in the low-salt environment. The polysaccharides in the temenos directly connected the polysaccharide chain or bridged the multivalent cations through the interaction of non-covalent bonds (Boyd & Chakrabarty, 1994) [44], which played a key role in temenos aggregation (Flemming et al., 1996) [45], thus forming a three-dimensional network in the temenos matrix. Proteins are involved in the construction of hydrophobic bonds in the EPS matrix (Jones et al., 2021) [46], maintaining the tertiary structure of proteins, which is conducive to their immobilization of macromolecules and particulate matter in the extracellular microenvironment, and their ability to provide sufficient low-molecular-weight nutrients for microbial aggregates. This further explains the reason for the enhanced stability of temenoses in high-salt environments.

In FQHC, bacteria were evenly distributed along the thickness of the temenos, while in SQHC, bacteria were enriched in the surface layer of the temenos (10 μm–20 μm), and were isolated from the concrete surface through deeper proteins and polysaccharides, thereby resisting the alkaline environment on the concrete surface. At the same time, the DNA of the EPS in SQHC was enriched at 30 μm–40 μm. It can be seen that, in a high-salinity environment, the temenoses accelerated the shedding and release of free spores, thereby promoting the survival of microorganisms, completing the temenos diffusion process, and accumulating microbial aggregates in new temenoses. Subsequently, sulfate-reducing bacteria infiltrated into the concrete cracks to produce biological acid, which destroyed the internal structure of the corroded concrete pipeline.

Even the content of polysaccharides in the samples tested in this study was low, but it can be clearly seen that the main distribution position of temenos polysaccharides in a high-salinity environment was significantly deeper than that in a low-salinity environment. The main distribution position of temenos polysaccharides indicates that the microbial aggregates in the deep part of the temenos in the high-salinity environment were highly active, and the free temenos structure was stable and could withstand the strong alkaline environment on the concrete surface (Mishra et al., 2023) [46]. This also provides an explanation for the partial shedding of temenoses in high-salinity environments and the rapid aggregation and adhesion of new spores to form new temenos aggregates (Ji et al., 2021) [47]. Research has shown that, in the process of the partial detachment and reaggregation of temenoses, lectin proteins in temenoses directly or indirectly cross-link polysaccharides through multivalent cations and play a key role (Saavedra et al., 2023) [48]. In addition, secreted bacterial polysaccharides play an important role in signal transduction and quorum sensing in temenoses (Araújo et al., 2019) [49].

In SQHC, there was a significant change in the lipid ratio at a depth of between about 30 μm and 43 μm. This may be attributed to the adaptive stress response of bacteria under high salt stress, in which carbohydrates are converted into lipids through specific enzymes to store energy (Shen et al., 2014) [50]. Previous studies have shown that growing bacterial cells tend to produce lipids when faced with environmental disturbances, which helps to reduce membrane fluidity, limit exchange, and save energy (Dubois-Brissonnet, 2018; Dubois-Brissonnet et al., 2016) [51,52]. This suggests that lipids may affect the temenos structure and bacterial survival in extreme environments.

## 4. Conclusions

This study utilized Avizo to analyze multi-fluorescent-labeled CLSM images, enhancing the 3D visualization modeling of EPSs. The model demonstrates that the temenos components are primarily concentrated in the central area of the temenos. A high salinity has a considerable effect on the distribution of temenos components, and as the temenos enters the mature stage, it further diffuses, with functional proteins being distributed along the temenos periphery. These proteins facilitate the active transport of low-molecular-weight nutrients, enabling microbial aggregates to survive within the high-alkalinity environment of concrete samples. Meanwhile, parts of the temenos detach from the concrete surface, and the microbial aggregates migrate into concrete cracks, further intensifying deterioration. Moreover, microorganisms regulate the types and spatial locations of amino acids, polysaccharides, and lipids to create optimal conditions for temenos survival and dissemination. The rationality of this model was corroborated by SEM observations of the temenos. It was evident that the temenoses on the concrete samples had reached maturity, forming EPSs on the surface. Compared to low-salinity conditions, it was observed that a higher salinity increases temenos detachment. In high-salinity environments, microbial penetration into the concrete pores was promoted, accelerating the deterioration of the concrete materials. This threatens the safe operation of sewage pipeline systems.

## Figures and Tables

**Figure 1 microorganisms-13-01452-f001:**
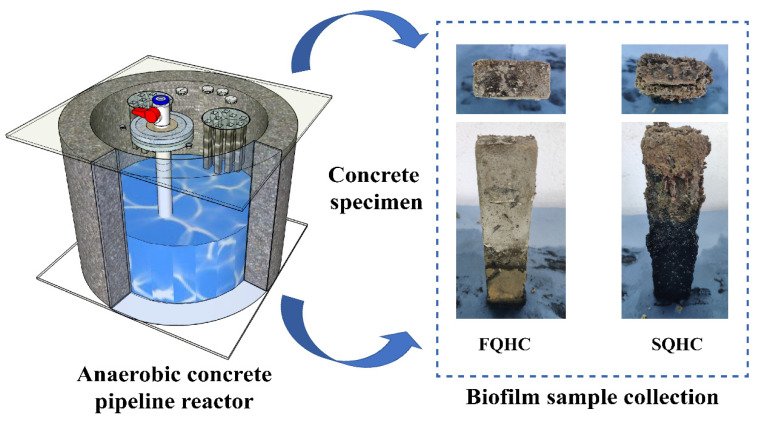
Anaerobic concrete pipeline and temenos sample.

**Figure 2 microorganisms-13-01452-f002:**
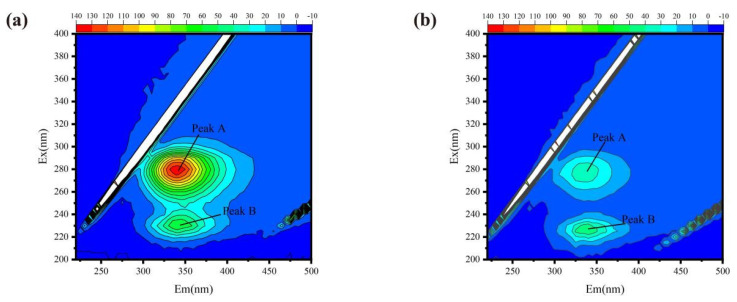
Three-dimensional EEM images: (**a**) FQHC; (**b**) SQHC.

**Figure 3 microorganisms-13-01452-f003:**
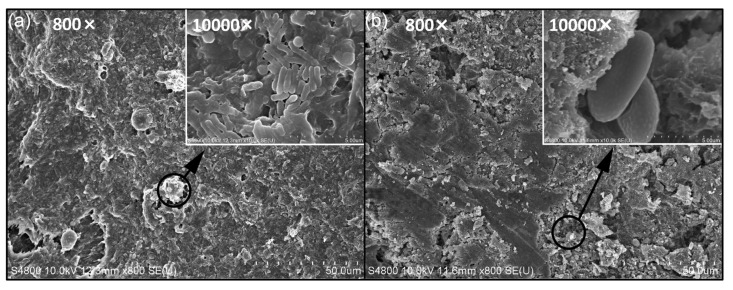
SEM image of temenos surface morphology: (**a**) FQHC; (**b**) SQHC.

**Figure 4 microorganisms-13-01452-f004:**
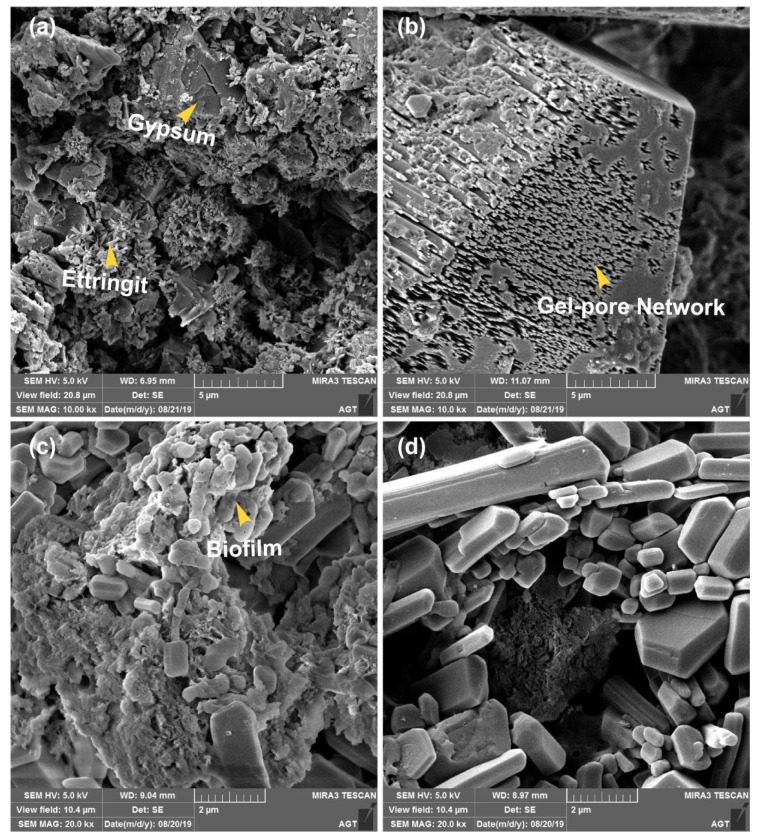
SEM images: (**a**) gypsum and ettringite on concrete surface; (**b**) biogenic acid erosion; (**c**) temenos attachment; and (**d**) temenos separation and diffusion.

**Figure 5 microorganisms-13-01452-f005:**
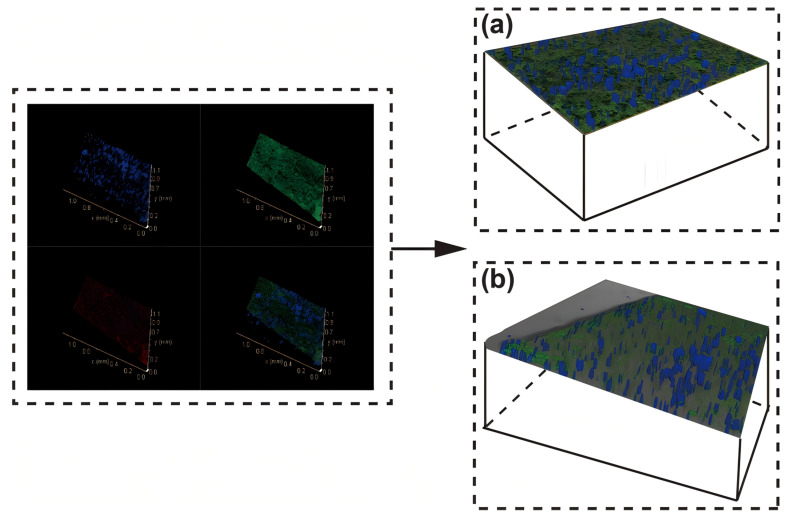
Three-dimensional structure model of temenos: (**a**) FQHC; (**b**) SQHC.

**Figure 6 microorganisms-13-01452-f006:**
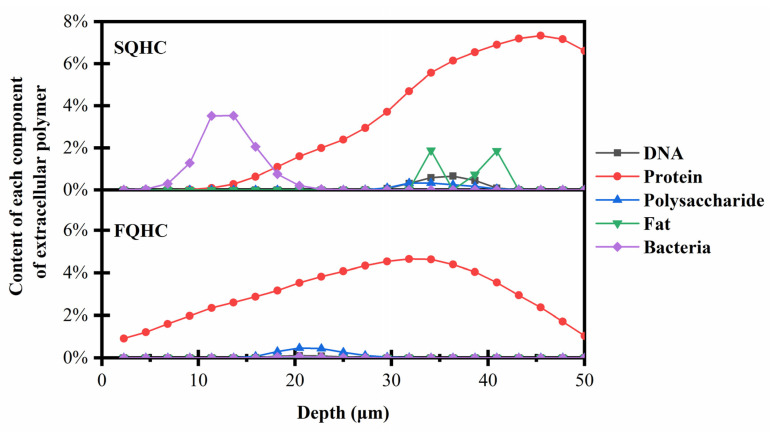
The relative proportion of temenos components in SQHC and FQHC changed with depth.

**Table 1 microorganisms-13-01452-t001:** Cross-talk check between stains at prescribed ex/em wavelengths.

Name of Fluorescein	Excitation Wavelength (nm)	Emission Wavelength (nm)
Calcofluor White	385, 395, 405	437, 440, 445
FITC	490, 494	520, 525
Nile Red	515, 555, 559	590, 640
Rhodamine	550	573
Propidium Iodide (PI)	(305), 536, 538	617

## Data Availability

The original contributions presented in this study are included in the article/Appendix A. Further inquiries can be directed to the corresponding author.

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
