# Peer review of "Reimagining Microbially Induced Concrete Deterioration: A Novel Approach Through Coupled Confocal Laser Scanning Microscope–Avizo Three-Dimensional Modeling of Biofilms"

_microorganisms, 2025, doi:10.3390/microorganisms13071452_

Round 1

Reviewer 1 Report

Comments and Suggestions for Authors

1.Note that biofilm is not a technically correct term although it is being used by many investigators (see: “Some thoughts about misconceptions surrounding the term ‘biofilm'”, Corrosion Engineering, Science and Technology, DOI: 10.1080/1478422X.2020.1774106, 2020.in addition to 

2. The term "corrosion" imples that there are anode and cathode in addition to electrolyte  and a metallic path. None of these exist in concrete. Therefore, it is technically correct to use "deterioration" instead of "corrosion" and thus it is more appropriate to use  microbiologically  influenced dteerioration (MID) instead of MICC.

I strongly advise to do these corrections.

Comments on the Quality of English Language

English quality is acceptable in my judgement.

Author Response

Comments 1:Note that biofilm is not a technically correct term although it is being used by many investigators (see: “Some thoughts about misconceptions surrounding the term ‘biofilm'”, Corrosion Engineering, Science and Technology, DOI: 10.1080/1478422X.2020.1774106, 2020.in addition to .

Response 1 : Thank you for pointing this out . We agree with this comment. Therefore,  We carefully read “Some thoughts about misconceptions surrounding the term ‘biofilm'.”Change the term "biofilm" to "temenos".These modifications have also been marked in yellow in the article.

Comments 2 : The term "corrosion" imples that there are anode and cathode in addition to electrolyte  and a metallic path. None of these exist in concrete. Therefore, it is technically correct to use "deterioration" instead of "corrosion" and thus it is more appropriate to use  microbiologically  influenced dteerioration (MID) instead of MICC.

Response 2 : Agree. we have accordingly revised. All corrosion in the article to deterioration, which has been marked in yellow in the article.

Reviewer 2 Report

Comments and Suggestions for Authors

The manuscript presents an innovative application of CLSM-AVIZO modeling to reimage Microbially Induced Concrete Corrosion (MICC). It explores the composition and content of extracellular polymeric substances (EPS), along with their functional roles under conditions of varying salinity. The study introduces a novel conceptual framework and demonstrates a promising new direction in the field. The writing is generally clear and the research approach is both rigorous and significant.

However, I recommend a revision of the manuscript before acceptance. Below are specific suggestions for improvement:

  1. EPS Collection and Analysis Procedures
    The description of biofilm collection and EPS component quantification in the Experimental section requires clarification. For instance, the phrase “centrifuge for 3 h” may be misleading—it is advisable to specify whether the sample was incubated for 3 hours prior to centrifugation. Additionally, the concentration of formaldehyde should be revised to indicate its final concentration after dilution with PBS.
  2. Microbial Characterization
    It is strongly recommended that the manuscript addresses potential bacterial strains present in the pipeline system. Including 16S rRNA analysis would substantially strengthen the identification of microbial communities involved in MICC.
  3. Material Composition Analysis
    The inclusion of Energy-Dispersive X-ray Spectroscopy (EDS) coupled with Scanning Electron Microscopy (SEM) is highly encouraged. This addition would provide critical insights into the atomic composition and distribution of materials affected by MICC.

Comments on the Quality of English Language

The English language writing is good 

Author Response

Comments1: EPS Collection and Analysis Procedures
The description of biofilm collection and EPS component quantification in the Experimental section requires clarification. For instance, the phrase “centrifuge for 3 h” may be misleading—it is advisable to specify whether the sample was incubated for 3 hours prior to centrifugation. Additionally, the concentration of formaldehyde should be revised to indicate its final concentration after dilution with PBS.

Response1:Thanks for the reminder, after communication with the team and repetition of the experiment, it was clearly stated that the final concentration of formaldehyde was 0.044% (v/v), calculated by calculation (0.06 mL formaldehyde / 50.06 mL total volume × 36.5% = 0.044%). Replace "3 hours of centrifugation" with "3 hours of incubation" and add an explanation of the operation during the incubation (intermittent shaking) to make it clear that the extraction process is chemical and not centrifugation alone. Thanks for the suggestion, we have revised it in part 2.5 of the article and marked it in orange font.

Comments2: Microbial Characterization
It is strongly recommended that the manuscript addresses potential bacterial strains present in the pipeline system. Including 16S rRNA analysis would substantially strengthen the identification of microbial communities involved in MICC.

Response2: Thank you for the suggestion. We added relevant content in the third part of the article, which is underlined. However, since the data has already been published, it is referenced at the end of the paragraph. We hope you understand. Finally, we would like to thank you again for your good advice. We genuinely respect you.

Comments3: Material Composition Analysis
The inclusion of Energy-Dispersive X-ray Spectroscopy (EDS) coupled with Scanning Electron Microscopy (SEM) is highly encouraged. This addition would provide critical insights into the atomic composition and distribution of materials affected by MICC.

Response3:Thank you for your suggestion, our team has published relevant content, after discussion, the relevant content will be effectively combined and supplemented in part 2.7 of the article, but since the data results have been published, so the citation, the changes are marked in purple, thank you again for your suggestion, this is a great help to us, I hope you are well, I and my team sincerely pay tribute to you.
